# Optimized FIR Filter Using Genetic Algorithms: A Case Study of ECG Signals Filter Optimization

Houssam Hamici [1], Awos Kanan [2] and Khalid Al-hammuri [3,*]

1 Department of Electrical Engineering, Princess Sumaya University for Technology, Amman 11941, Jordan; hou20218181@std.psut.edu.jo
2 Department of Computer Engineering, Princess Sumaya University for Technology, Amman 11941, Jordan; a.kanan@psut.edu.jo
3 Department of Electrical and Computer Engineering, University of Victoria, Victoria, BC V8W 2Y2, Canada
* Correspondence: khalidalhammuri@uvic.ca

**Abstract:** The advancement in technology and the availability of specialized digital signal processing chips have made digital filter design and implementation more feasible in a variety of fields, including biomedical engineering. This paper makes two key contributions. First, it uses a genetic algorithm to optimize the coefficients of finite impulse response (FIR) filters. Second, it conducts a case study on using genetic algorithms to optimize FIR filters for electrocardiogram (ECG) biomedical signal noise removal. The goal of the proposed filter design approach is to achieve the desired signal bandwidth while minimizing the side lobe level and eliminating unwanted signals using a genetic algorithm. The results of a comprehensive analysis show that the genetic algorithm-based filter is more effective than conventional filter designs in terms of noise removal efficiency.

**Keywords:** genetic algorithms; digital filters; metaheuristic optimization; finite impulse response (FIR); side lobe level (SLL)





## 1. Introduction

### 1.1. Problem Statement

Electrocardiography (ECG) is one of the common biomedical signals that are used to predict and assess heart health status that results from weak heart muscle myocardial infarction, heart attacks, or other related diseases. To obtain optimal results, ECG signals must be analyzed by using Holter monitors to obtain a real-time analysis. While this is more efficient, it requires a special algorithm to handle the computational complexity of huge data, involving noise removal, analysis, and post-processing.

Digital filters are widely used in many different fields to remove noise, to split multiple signals [1], for missing data imputation [2], and to enhance the quality of received signals [3,4]. In recent years, genetic algorithms (GA) have been increasingly used to optimize the design of digital filters because they can handle the issue of computational complexity and big data.

GA [5] is a metaheuristic optimization algorithm that can be used to find optimal, or near optimal, solutions to problems that are too computationally expensive and are difficult to solve using traditional methods. In the case of digital filter optimization, the goal is to find a set of coefficients that optimizes the performance of the filter using some performance metric.

Electrocardiography (ECG) is a well-known and reliable tool that is widely used in the medical field to analyze and diagnose the condition of the heart. This analysis is very crucial for the early detection of heart diseases. ECG signals are very weak signals and, hence, susceptible to disturbances in the measurement environment. Using digital filters to

remove unwanted parts of these signals is critical for the performance and accuracy of the algorithms and software tools that are used to detect abnormal heart conditions [6].

The GA approach to digital filter optimization has several advantages over traditional methods [7,8]. First, it is not limited to finding local minima, which can be a problem with traditional methods. Second, it can be used to find solutions to problems that are too complex for traditional methods. Third, it is relatively easy to implement. This approach is a promising technique that has the potential to significantly enhance the design of digital filters used in different fields, including biomedical engineering.

### 1.2. Contribution Statement

In this paper, a GA-based digital filter is proposed to optimize the coefficients of FIR filters for ECG signals. The search space for the filter is defined by 20 coefficients. Finding the optimal values of these coefficients is practically extremely large due to the high non-linearity of the frequency response of the filters.

The actual coefficients of the filter are mapped to the genes of a chromosome. Each coefficient represents a gene, while the whole coefficients represent a chromosome. The genetic algorithm works by iteratively generating new populations of filters, each of which is a slight variation of the previous population. The filters in each population are evaluated based on their SLL, and the best filters are selected to create the next generation. This process continues until a filter with a sufficiently low SLL is found.

The objective, or fitness, function to be optimized is the minimization of the side lobe level (SLL) [9]. The side lobe level is a measure of the noise that is produced by the filter outside of its passband. A low side lobe level indicates that the filter is good at removing noise. The genetic algorithm works by iteratively generating new populations of filters, each of which is a slight variation of the previous population. The filters in each population are evaluated based on their side lobe level, and the best filters are selected to create the next generation. This process continues until a filter with a sufficiently low side lobe level is found.

The remainder of the paper is organized as follows. Section 2 presents the related work. Section 3 explains the research methodology. Section 4 highlights and compares different options of the proposed methodology. Moreover, Section 5 discusses the results for the case study of using GA to filter ECG signals. Finally, Section 6 concludes the paper.

### 2. Related Work

Metaheuristic optimization algorithms such as (GA) [10], ant colony optimization (ACO) [11], and chemical reaction optimization (CRO) [12] are used to solve different optimization problems. These algorithms have been extensively used in recent years to implement digital filters in a wide range of applications [13].

The authors in [14] proposed a comprehensive fitness function for an infinite impulse response (IIR) digital filter design with six terms, including a low delay parameter. Low delay filters are preferred since they achieve quasi-real-time processing. Other parameters can be included in the fitness function to meet linear phase, minimum phase, and stability constraints.

The authors in [15] proposed a method to detect breast malignancies. The filter bank that has been used to obtain the filter response is the Maximum Response Eight Filter Bank. (GA) and linear discriminant analysis have been used for feature selection and feature reduction, respectively.

An optimal fourth-order Butterworth active low-pass filter design by the hybrid intensified current search (HICuS) algorithm was proposed in [16]. With AI-based modern optimization, the HICuS algorithm is claimed to be one of the most efficient metaheuristic optimizers.

The authors in [17] presented a differential evolution algorithm that uses a combination of rectangular and polar coordinates. The algorithm was developed using thirteen

commonly used numerical optimization test functions. The algorithm was then applied to design an infinite impulse response (IIR) digital filter.

A digital filter design problem that involves multiple, often conflicting, filter parameters has been presented in [18]. Hence, an optimization algorithm is needed. A low-pass finite impulse response (LPFIR) filter optimal design has been achieved using particle swarm optimization (PSO) and dynamic adjustable PSO (DAPSO). Other recent works in [19–23] presented the design of FIR with variable multiple stop band, coefficients optimization in adaptive equalization, and FIR design using arithmetic optimization and (GA).

Similarly, there are different techniques used to denoise the ECG signal. The denoising techniques include empirical mode decomposition (EMD), ensemble empirical mode decomposition (EEMD), and discrete wavelet transform (DWT) [24]. Denoising ECG signals were also explored in [25] by using time–frequency techniques.

At the same time, there are different techniques used to filter and analyze electrical signals extracted from humans, like electromyography (EMG), as cited in [26]. These methods are complete ensemble empirical mode decomposition with adaptive noise (CEEMDAN) and ensemble empirical mode decomposition (EEMD). In this technique, the signals are analyzed by time–frequency techniques such as adaptive optimal kernel (AOK) and Choi–Williams.

Compared to the conventional techniques used to design digital filters, the proposed genetic algorithm technique provides better exploration of the problem search space to avoid getting stuck with a local optimum. It also provides easy and straightforward encoding of the filter optimization problem.

## 3. Methodology

The research methodology is described in four stages as depicted in Figure 1. Stage one is FIR filter design and coefficient generation. Stage two uses (GA) to map the filter coefficient. Stage three is to predict the best solution of the filter coefficient using a genetic algorithm. Stage four evaluates the (GA) prediction using impulse response.

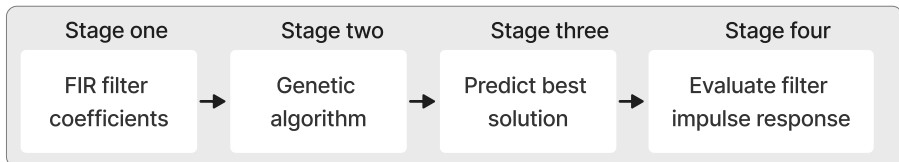

**Figure 1.** Research methodology flow chart.

Stage one (FIR filter design): The filtering process uses a direct-form discrete-time FIR filter of order $M$. The top part is an $M$-stage delay line with $M + 1$ taps shown in Figure 2. Each unit delay is a $z^{-1}$ operator in $Z$-transform notation. For a causal discrete-time FIR filter of order $M$, each value of the output sequence is a weighted sum of the most recent input values as in Equation (1).

$$y[n] = \sum_{i=0}^{M} b_i x[n - i] \tag{1}$$

where $x[n]$ and $y[n]$ are the discrete input and output signals, respectively. $M$ is the filter order, and $b_i$ is the value of the impulse response at the $i$th instant for $M$ of an $M$-order FIR filter.

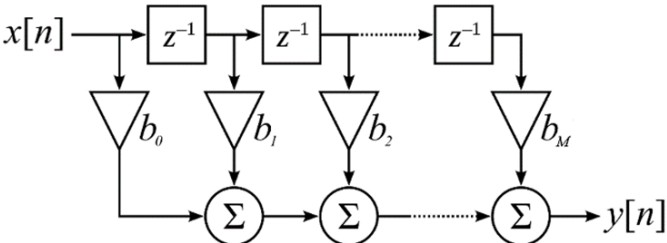

**Figure 2.** Structure of a direct-form discrete-time FIR of order *M*.

The frequency response $H(z)$ is expressed in terms of the *Z*-transform transfer function as shown in Equation (2):

$$H(z) = \frac{Y(z)}{X(z)} = \frac{\sum_{k=0}^{M} b_k z^{-k}}{1 + \sum_{k=1}^{N} a_k z^{-k}} \qquad (2)$$

Ideally, we would like to have *M* negligible for small computations with a target frequency response provided by the ideal low-pass filter as follows in Equation (3):

$$H(\omega) = \begin{cases} 1, if \ |\omega| \leq |\omega_c| \\ 0, if \ \omega_c \leq |\omega| \leq \pi \end{cases} \qquad (3)$$

Stage two (map and process FIR coefficients using GA): The general structure of GA is depicted in Figure 3. The process started by choosing a sample of an initial population of 20 chromosomes, each one containing 20 genes, which was created randomly in the range [0, 1]. The genes represent the filter coefficients. For this research, the results are optimized for 20 coefficients to create a balance between computation and filter enhancement. However, the number of filter coefficients can be altered based on the application and nature of the data and their associated noise.

Following the selection of filter coefficients, the initial set was passed to a fitness function for frequency response evaluation and side lobe level (SLL) estimation. The selection is completed for the minimum SLL. If the condition of the minimum SSL level does not meet the desired threshold, the GA selects new genes or filter coefficient randomly using crossover and mutation, re-evaluates the fitness function SSL, and repeats the previous procedures.

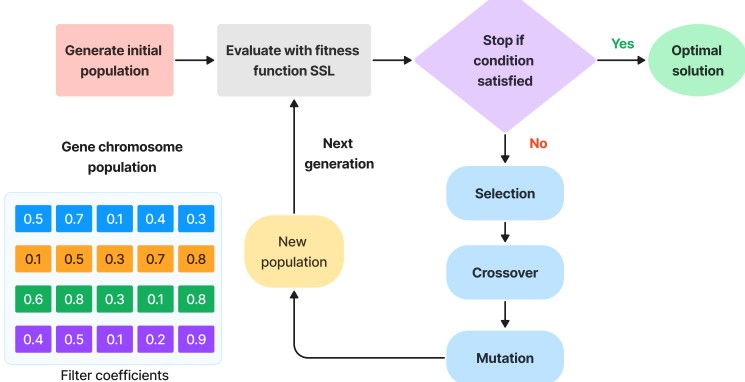

**Figure 3.** General structure of a (GA) for digital filter applications.

Stage three (predict the best solution): The mapped filter coefficients are filtered out by using GA to select the best solution among the coefficients. Figure 4 depicts 20 frequency responses for the population and emphasizes the best solution shown in red with a thick solid line. Figure 5 illustrates the frequency response in magnitude for the best solution extracted from the initial population. The spectrum was extracted using fast

Fourier transform [27]. The coefficients of the best solution are shown below. The SLL is shown in Figure 5 with −16.1660 dB.

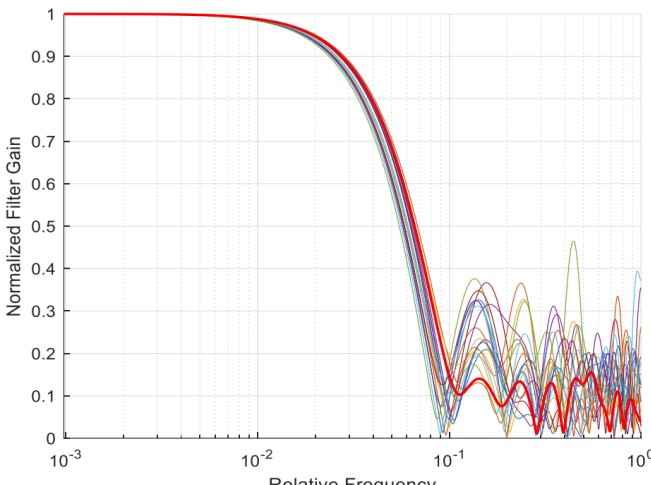

**Figure 4.** A normalized filter gain vs relative frequency of a population for 20 filters (chromosomes). Each filter is visualized in differnt colours. The best filter is shown in red with minimum side lobe level (SLL). Relative or normalized frequency is in logarithmic scale. The filter order is $N^{-1} = 19$.

Stage four (evaluate the best solution): Finally, the best solution is evaluated using the impulse response to select the minimum SSL. This step is important to make sure the best solution is optimized in the desired frequency bandwidth and is at the extreme minimum level for other frequencies. The GA has some randomness in the process, and this step is essential as a final quality control measure for choosing the best solution. Figure 5 depicts the impulse response coefficient of the best solution extracted from the initial population provided by $b[n]$ coefficients. In this example, filter coeffs ($b[n]$) = [0.5009 0.4457 0.6175 0.5180 0.4395 0.8174 0.7331 0.9891 0.8428 0.7717 0.6087 0.9935 0.4332 0.3967 0.4869 0.6107 0.3800 0.0383 0.8333 0.7669]. The best SLL from the initial population is −16.1660 dB .

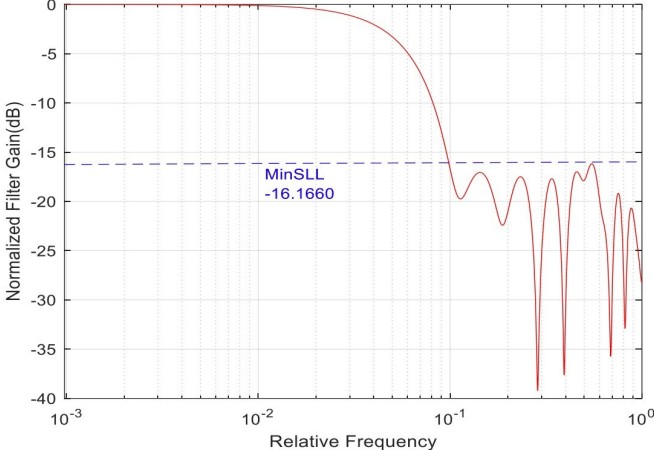

**Figure 5.** The the best filter based on the minimum SLL selected from the initial population. The SLL is at −16.166 dB. Relative or normalized frequency is in logarithmic scale. Filter order is ($N^{-1}$) = 19.

## 4. Comparative Analysis of the GA Algorithm Options for FIR Filter Optimization

The analysis of the GA algorithm is formulated as an optimization problem. The core of an optimization problem is based on the minimization of a fitness function. The latter has to be set meticulously to achieve our objective of reaching a minimum SSL. The fitness function has input variables provided by the filter coefficients and the resolution of the

frequency response of the filter. The objective is to minimize the maximum of the SLL as an output. This output is supplied to the (GA) for minimization.

Detailed results and analysis for the study are listed and discussed in Sections 4.1–4.4, in addition to detailed information in Appendices A and B. These sections summarize the comprehensive study options. Each section shows results regarding using optimization solver functions with sub-functions for each GA algorithm option. The results are provided by the SLL and the filter coefficients, along with graphs on convergence and frequency response. The GA is set with different options in order to study the effect of each parameter on the result of the minimization. The GA uses other options provided by the solver.

The fitness function that relies on estimating the maximum of the SLL of the normalized discrete frequency response is discussed in the following sections. For each solver option from the four alternatives, a graph compares the four SLL cases, and another graph compares the convergence. The GA results are compared with the conventional FIR results generated by Matlab V.R2022n, where the seed of the random number in the Matlab generator is fixed.

### 4.1. Algorithm Settings Results and Analysis

The study of algorithm settings has two categories: crossover function and mutation function. The crossover function has two options: heuristic and crossover two-point. At the same time, the mutation options are adapt-feasible and Gaussian.

Figure 6 shows the gain with respect to normalized frequency for cases (1–4); see Appendix A. The lowest performance in Figure 6 is provided by the heuristic function. Crossover two-point, Gaussian mutation and adapt-feasible mutation demonstrate similar performance up to a 1 dB margin.

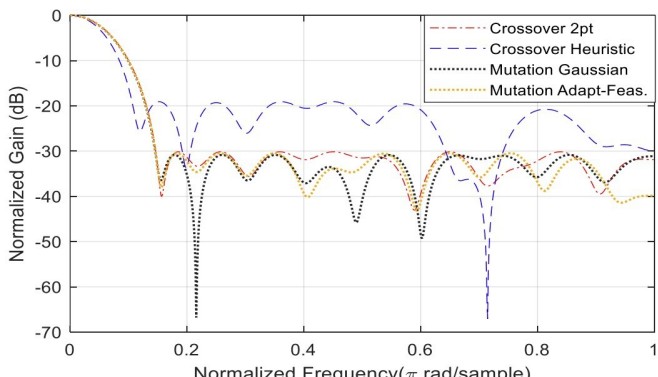

**Figure 6.** Minimization of the SLL. The magnitude spectrum shows two algorithm settings, each with two functions, as described in the legend.

At the same time, for the above-mentioned four cases, the convergence speeds are different, as shown in Figure 7. The SLL and the filter coefficients are listed for each case; see Appendix A. The crossover heuristic has the best performance since it converges in 71 generations, followed by crossover two-point with 193 generations, and then comes mutation with adapt-feasible, followed by Gaussian mutation with 235 generations and 326 generations, respectively.

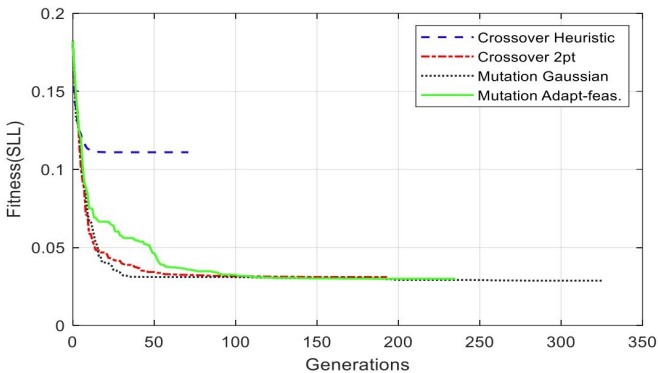

**Figure 7.** Illustration of algorithm setting and analysis four categories convergence speed of the SLL minimization. Four algorithm-setting categories are depicted in the figure.

The results of various options of algorithm settings are listed in Table 1. The population size is 200 for all cases. By analyzing Figures 6 and 7, we can observe that the optimal solution is provided by crossover with a two-point function since it achieves an SLL close to the best solution with moderate convergence.

**Table 1.** Algorithm setting evaluation results.

| Alg. Setting | No. | Sub-Function Solver Setting | SSL (dB) | Gen. No. Convergence |
|---|---|---|---|---|
| Crossover | 1 | Heuristic | −19.0941 | 71 |
| Crossover | 2 | Two-point | −30.1599 | 193 |
| Mutation | 1 | Adapt-Feasible | −30.4694 | 235 |
| Mutation | 2 | Gaussian | −30.8327 | 236 |

Further analysis and experimental results for mutation and crossover are presented in Appendix B.

*4.2. Population Settings Results and Analysis*

The population setting has two solver options: max gens and stall time limit. Max gens were evaluated using two sub-functions, settings 100 and 200, while the stall time sub-functions are set as 2 and 4 s. Table 2 summarizes the results for various options on the configurations of the GA population setting category. Each solver option is studied, with two sub-functions each. The SSL values are presented in dB.

**Table 2.** Population setting evaluation results.

| Solver Option | No. | Sub-Function Solver Setting | SSL (dB) | Gen. No. Convergence |
|---|---|---|---|---|
| Max gens | 1 | 100 | −30.0219 | 100 |
| Max gens | 2 | 200 | −30.1599 | 193 |
| Stall time limit | 1 | 2 s | −30.1599 | 193 |
| Stall time limit | 2 | 4 s | −30.1599 | 193 |

In Figure 8, when max generation is set to 100, the SLL stops at −30.0219, while the SLL stops at SLL of −30.1599 dB at generation 193 before the max generation of 200. For a stall time limit of 2 and 4 s, the SLL converges to the same value with the same filter coefficients.

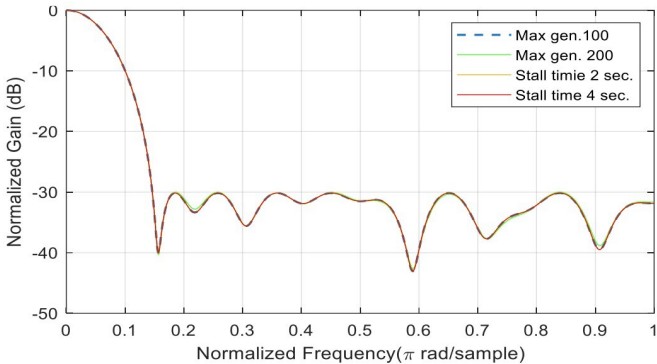

**Figure 8.** Minimization of the SLL (max gen. 100 and 200).

Figure 9 shows the convergence speed of the SLL. By analyzing Figures 8 and 9, we can observe that increasing the number of generations or stall time limit will not improve the SLL if the fitness converges before reaching the set values.

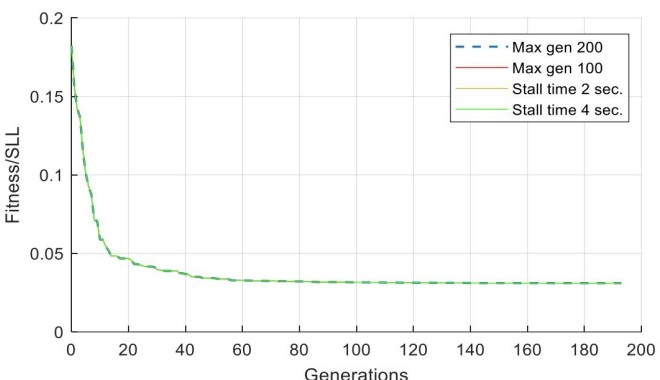

**Figure 9.** Convergence speed of the SLL minimization. For max generations of 100, the fitness stops exactly at 100 generations, while, for the three other cases, the generations stop at 193 generations. The lines for the four categories are not clear because they have the same convergence in this case. Please refer to Table 2 for detailed numbers.

### 4.3. Runtime Limits Results and Analysis

Runtime limits have two categories: population size and initial population. Population size has two sub-function solver settings: 100 and 300. In contrast, the initial population has two solver settings: $x_0$ and $x_1$; see Appendix A for more details. The SSL convergence of the number of population settings category is depicted in Figure 10 with respect to the population sample of different generations numbering 202, 235, 253, and 274, as listed in Table 3. The initial values for $x_1$ are population size 300, population size 100, and initial $x_0$, respectively. Changing the population size or initial conditions will shift the convergence speed. Increasing the population size will decrease the convergence iterations.

**Table 3.** Runtime limits.

| Pop. Setting | No. | Sub-Function Solver Setting | SSL (dB) | Gen. No. Convergence |
|---|---|---|---|---|
| Pop. Size | 1 | 100 | −30.7976 | 253 |
| Pop. Size | 2 | 300 | −31.5810 | 235 |
| Initial. Pop | 1 | Initial Pop. ($x_0$) | −31.3811 | 274 |
| Initial. Pop | 2 | Initial Pop. ($x_1$) | −31.3141 | 202 |

Figure 11 shows that the default value of the GA is 200 chromosomes. When the population is set to 300, the SLL is almost constant when the initial condition is varied. It changes from −31.3811 dB to −31.3141 dB, with a difference of 0.067 dB.

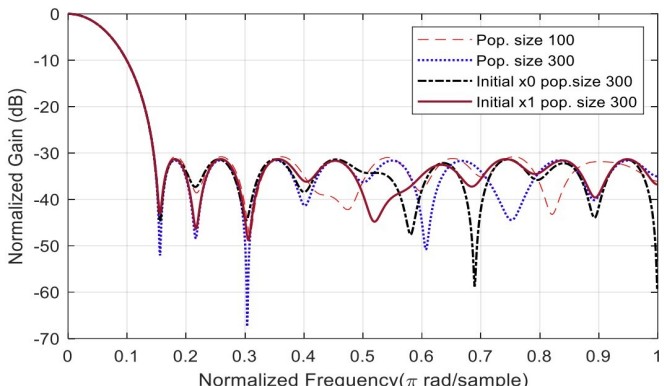

**Figure 10.** Minimization of the SLL. For population size increasing from 100 to 300, the SLL decreases from −30.7976 dB to −31.5810 dB, with a slight difference of less than 1 dB.

By analyzing Figures 10 and 11, we can observe the effect of changing the population size and the change in the initial condition on the algorithm convergence for different setting categories.

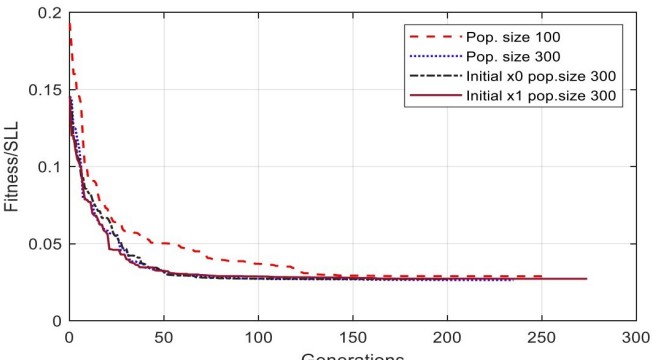

**Figure 11.** Convergence speed of the SLL minimization.

*4.4. Tolerance Results and Analysis*

The tolerance option has two categories: function tolerance and fitness limit. The function tolerance solver setting is set to the $e^x$, and the fitness limit study selects 0.01 and 0.001 as two different decimal sub-function solvers. Table 4 summarizes the tolerance option setting of the GA algorithm. The table also lists different solver options and different generation numbers. Varying the fitness limit from 0.01 to 0.05 changes the SLL from −30.1599 dB to −26.3296 dB; see Figure 12 for more details. For instance, $10^x$ corresponds to $10^x$ (base 10, not base e = 2.71). For the convergence speed, see Figure 13. When the function tolerance decreases from $10^{-5}$ to $10^{-4}$, the convergence decreases from 148 gen. to 92 gen. When the fitness limit varies from 0.01 to 0.05, the convergence gains high speed from 193 generations to 14 generations.

**Table 4.** Tolerance evaluation results.

| Solver. Options | No. | Sub-Function Solver Setting | SSL (dB) | Gen. No. Convergence |
|---|---|---|---|---|
| Fun. tolerance | 1 | $10^{-4}$ | $-29.9617$ | 92 |
| Fun. tolerance | 2 | $10^{-5}$ | $-30.1490$ | 148 |
| Fitness limit | 1 | 0.01 | $-26.3296$ | 14 |
| Fitness limit | 2 | 0.001 | $-30.1599$ | 193 |

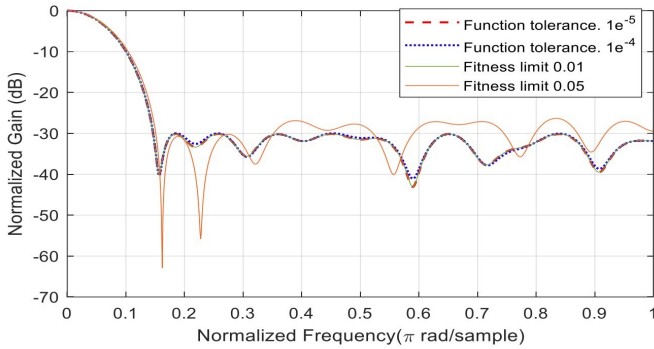

**Figure 12.** Minimization of the SLL, changing the function tolerance from $10^{-5}$ to $10^{-4}$, shifting the SLL from $-30.1490$ dB to $-29.9617$ dB with faster convergence. Where the base e = 10 in the figure.

By analyzing Figures 12 and 13, we can observe the effect of changing the function tolerance and the fitness limit on both the SLL and the convergence speed (number of generations required for the stop criteria), as depicted in the figures' captions.

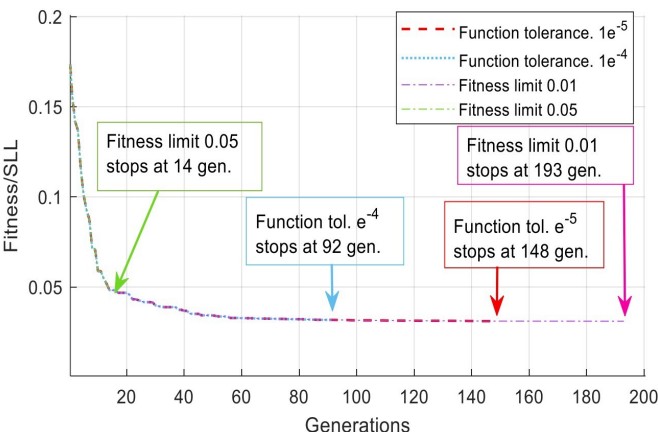

**Figure 13.** Convergence speed of the SLL minimization of different tolerance options. Where the base e = 10 in the figure.

## 5. ECG Filtering Results Discussion

This section presents a case study for the optimized FIR filter coefficients using GA. The validation of the proposed algorithm is performed on an ECG signal where an existing noise is to be removed, and the comparison of both time and frequency signals is carried out. The signal was then analyzed to assess the quality of the filtered signal. Comprehensive results are shown below in different figures.

Figure 14 depicts the ideal ECG signal for a normal person. The figure shows the main components of the ECG, which are the P-wave, QRS complex, and T-wave. The ideal signal acts as a reference while removing the noise to make sure all the components are preserved.

The case study discusses a noisy ECG signal contaminated with 60 Hz noise and other artifacts. The study removes the artifacts from a noisy ECG signal using GA. The 60 Hz noise from the power line is the common source of interference that affects the ECG signal.

It has a considerable effect as its amplitude is significant to interfere with the low-amplitude ECG signal, which is in the range of Millivolts (mV) in most cases. Furthermore, there are different types of artifacts and interference that affect ECG signals, like body movement and external electromagnetic waves that are generated from other devices, mainly from medical imaging devices at hospitals.

As illustrated in Figure 15, the noise effect is significant regarding P-wave and T-wave as they have lower amplitude than the QRS complex of the ECG. Furthermore, there is an artifact at the beginning of the signal; this artifact is typically generated from motion or improper electrode attachments. The noise in the signal misleads the medical doctor while identifying the patient's case.

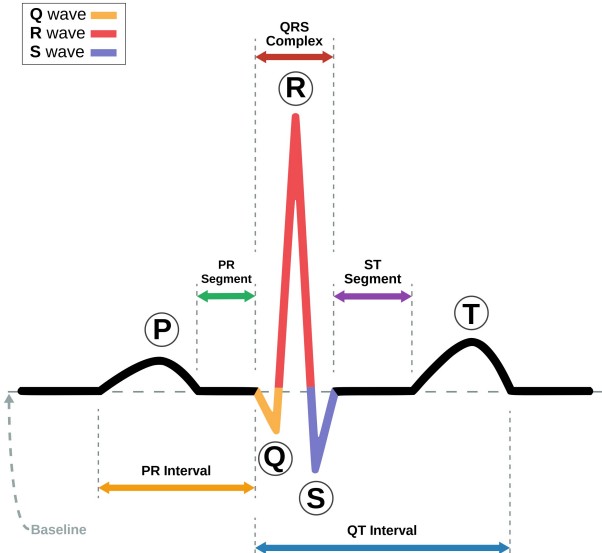

**Figure 14.** Ideal ECG signal for a normal person.

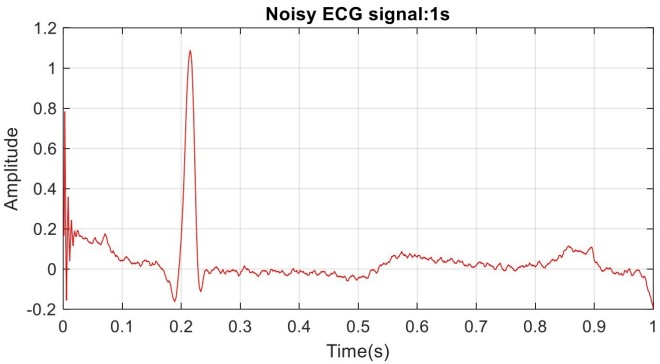

**Figure 15.** Example of one second of the noisy ECG signal.

To analyze the source of the noise on the complex structure of the ECG signal, we need to analyze the magnitude and frequency response of the spectrum. This is important to identify the signal that has a dominant spectrum over the typical range of ECG.

Figure 16 is the magnitude spectrum displaying the level of noise in dB with respect to the frequency in Hz. Visualizing the noisy signal in the frequency domain helps to identify the dominant noisy signal on the original data.

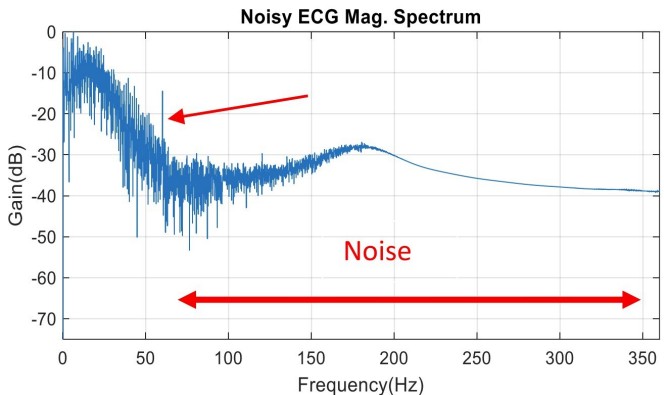

**Figure 16.** The magnitude spectrum of the noisy ECG signal. The ECG signal is contaminated with a 60 Hz signal from the power line. The sampling rate is 720 samples/s. Half sampling is at 360 Hz.

Figure 17 is the spectrum of the filtered signal. The filtered ECG uses GA with the Gaussian mutation category, which shows an attenuation of noise with more than 30 dB. At half sampling rate, the signal is at −70 dB. The interference signal of 60 Hz is attenuated with more than 30 dB. The analysis is for case 3; see Appendix A for details.

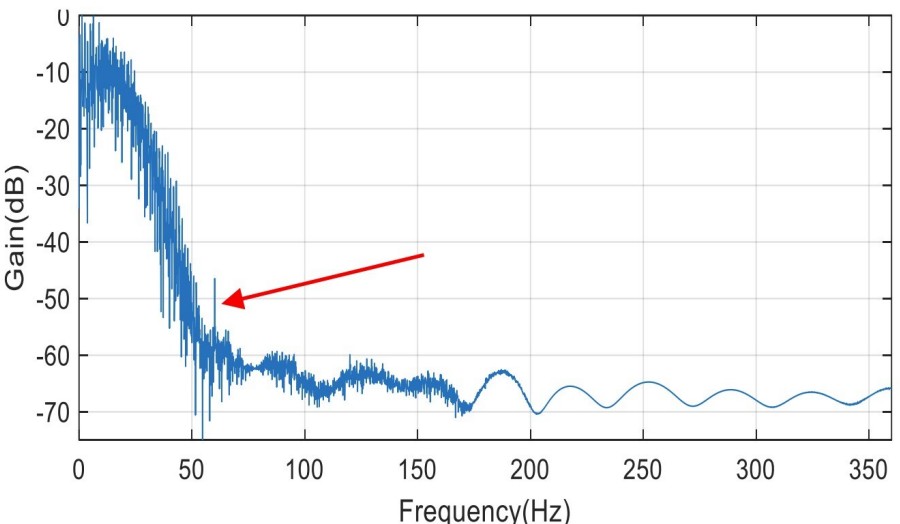

**Figure 17.** Filtered ECG spectrum magnitude using GA Gaussian mutation.

The final filtered ECG is shown in Figure 18 with respect to the time domain. Comparing Figures 15 and 18 shows the excellent filtering behaviour of the Gaussian mutation *GA* filter. The ECG signal in Figure 18 may be different from the ideal ECG in Figure 14 as the real recording has some variations for normal cases between different patients.

Figure 19 shows the difference between the GA and standard FIR filter frequency response. The figure shows the standard FIR attenuation gain as −40 dB compared to −31 dB for the *GA*. The GA FIR presents a sharp passband relative to the standard FIR magnitude for the same order, even though the relative cutoff frequency is set to 0.01 (1%).

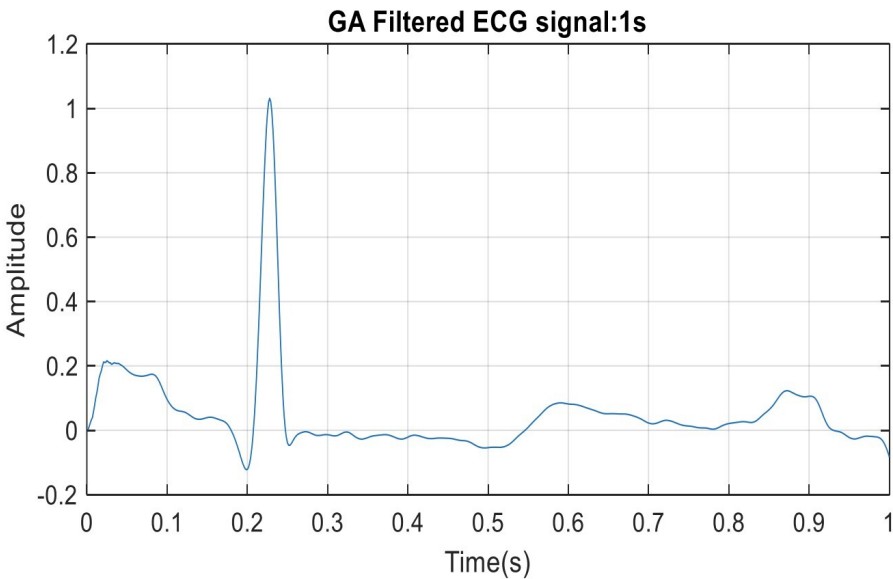

**Figure 18.** One second at the beginning of the filtered signal showing complete removal of the artifact, 60 Hz power signal interference with smoothed ECG signal using Gaussian mutation.

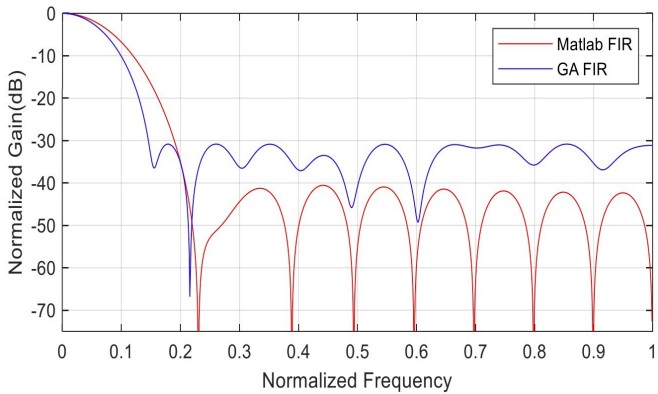

**Figure 19.** Comparison between GA FIR filter and conventional FIR for low-pass filters generated by Matlab.

## 6. Conclusions

In this study, a GA is used to optimize and comprehensively analyze the synthesis of digital low-pass filters. Sixteen cases are analyzed, and the results are shown to demonstrate the effectiveness of the method. The fitness function is derived from the frequency response of the finite impulse response low-pass filter. The inputs of the fitness are the coefficients of the filter, which start with a random sequence, in addition to the frequency resolution of the frequency response. The output is the side lobe level that has to be minimized for the optimal filter synthesis. Different GA options have been tuned during the optimization process to enhance the performance of the filter for a noisy real ECG recording. The obtained filter shows enhanced results in suppressing different kinds of noise.

For future work, we suggest using deep learning algorithms to optimize filter design. In the case of huge and complex datasets, a transformer-based model is recommended to handle enormous samples as they can be processed in parallel.

**Author Contributions:** Conceptualization, H.H. and A.K.; methodology, H.H. and A.K.; software, H.H. and K.A.-h.; validation, H.H. and K.A.-h.; formal analysis, H.H. and K.A.-h.; investigation, H.H., A.K. and K.A.-h.; resources, H.H. and K.A.-h.; data curation, H.H. and K.A.-h.; writing—original draft preparation, H.H. and A.K.; writing—review and editing, A.K. and K.A.-h.; visualization, H.H. and K.A.-h.; supervision, A.K.; project administration, A.K. and K.A.-h. All authors have read and agreed to the published version of the manuscript.

**Funding:** This research received no external funding.

**Institutional Review Board Statement:** Not applicable.

**Informed Consent Statement:** Not applicable.

**Data Availability Statement:** ECG signals database is publicly available at http://www.physionet.org/physiobank/database/mitdb/ (accessed on 20 June 2023).

**Conflicts of Interest:** The authors declare no conflict of interest.

## Abbreviations

The following abbreviations are used in this manuscript:

| | |
|---|---|
| ACO | Ant colony optimization |
| Alg. | Algorithm |
| CRO | Chemical reaction optimization |
| DAPSO | Dynamic adjustable particle swarm optimization |
| DSP | Digital signal processing |
| dB | Decibel unit |
| ECG | Electrocardiogram |
| EEG | Electroencephalogram |
| FIR | Finite impulse response |
| *GA* | Genetic algorithm |
| Gen. | Generation |
| IIR | Infinite impulse response |
| PSO | Particle swarm optimization |
| *SLL* | Side lobe level |

## Appendix A. Comparative Analysis of Different *GA* Categories and Options

This section presents the results of 16 cases, as listed in Table A1. The results were analyzed using different GA algorithms and different categories. For each option, there are selected sub-function solvers. The filter coefficient and the SSL are also listed to show the best solution among different options.

**Table A1.** Comparison between different methodologies.

| Case No. | GA Algorithm Option | Category | Sub-Function Solver Setting | SSL (dB) | Filter Coefficients |
|---|---|---|---|---|---|
| 1 | Algorithm Setting | Crossover function | Heuristic | −19.0941 | [0.1728 0.5336 0.5986 0.2363 0.4960 0.7534 0.6947 0.5135 0.4703 0.6803 0.6439 0.6650 0.7603 0.2939 0.4812 0.3335 0.4335 0.3923 0.3140 0.3530] |
| 2 | Algorithm Setting | Crossover function | Crossover two-point | −30.1599 | [0.1119 0.1180 0.1937 0.3267 0.3476 0.4794 0.5589 0.6952 0.7050 0.7677 0.8614 0.7511 0.7524 0.7646 0.6211 0.5923 0.4037 0.4056 0.3399 0.2514] |
| 3 | Algorithm Setting | Mutation Function | Gaussian mutation | −30.8327 | [0.1359 0.2439 0.2813 0.3658 0.4837 0.6359 0.6791 0.8025 0.8434 0.8723 0.9373 0.9739 0.8859 0.8013 0.6949 0.5921 0.5364 0.4193 0.2180 0.3143] |
| 4 | Algorithm Setting | Mutation Function | Adapt-feasible mutation | −30.4694 | [0.2063 0.2968 0.3638 0.5322 0.6796 0.7954 0.8321 0.9273 1.0000 1.0000 0.9863 0.9579 0.9579 0.8361 0.6728 0.4515 0.5123 0.2934 0.2521 0.2428] |
| 5 | Population Setting | Population Size | Size = 100 | −30.7976 | [0.1575 0.2661 0.2467 0.4623 0.5886 0.5737 0.7992 0.9097 0.9834 0.9999 0.9954 1.0000 0.9933 0.8660 0.7521 0.6720 0.5120 0.4924 0.2738 0.2611] |
| 6 | Population Setting | Population Size | Size = 300 | −31.5810 | [0.1916 0.2769 0.3671 0.3364 0.6227 0.6609 0.7728 0.8591 0.9155 0.9637 0.9567 0.9225 0.8709 0.7921 0.6803 0.5446 0.5021 0.3453 0.2449 0.2105] |
| 7 | Population Setting | Initial Population | value $x_0$ = [0.5425 0.1422 0.3733 0.6741 0.4418 0.4340 0.6178 0.5131 0.6504 0.6010 0.8052 0.5216 0.9086 0.3192 0.0905 0.3007 0.1140 0.8287 0.0469 0.6263] | −31.3811 | [0.3034 0.2449 0.3330 0.4338 0.5682 0.7429 0.8100 0.8999 0.9730 0.9947 0.9979 0.9346 0.8853 0.8487 0.6816 0.5995 0.5000 0.3849 0.2243 0.1835] |
| 8 | Population Setting | Initial Population | value $x_1$ = [0.5476 0.8193 0.1989 0.8569 0.3517 0.7546 0.2960 0.8839 0.3255 0.1650 0.3925 0.0935 0.8211 0.1512 0.3841 0.9443 0.9876 0.4563 0.8261 0.2514] | −31.3141 | [0.2438 0.2903 0.4232 0.5260 0.5379 0.7947 0.8925 0.8748 0.9824 1.0000 0.9999 0.9441 0.8710 0.7934 0.6194 0.5883 0.4087 0.3561 0.2031 0.1954] |
| 9 | Runtime Limits | Max Generations | value = 100 | −30.0219 | [0.1087 0.1176 0.1907 0.3214 0.3442 0.4764 0.5565 0.6933 0.7032 0.7667 0.8613 0.7482 0.7511 0.7637 0.6208 0.5923 0.4035 0.4112 0.3396 0.2504] |
| 10 | Runtime Limits | Max Generations | value = 200 | −30.1599 | [0.1119 0.1180 0.1937 0.3267 0.3476 0.4794 0.5589 0.6952 0.7050 0.7677 0.8614 0.7511 0.7524 0.7646 0.6211 0.5923 0.4037 0.4056 0.3399 0.2514] |
| 11 | Runtime Limits | Stall Time Limit | value (T) = 2 s | −30.1599 | [0.1119 0.1180 0.1937 0.3267 0.3476 0.4794 0.5589 0.6952 0.7050 0.7677 0.8614 0.7511 0.7524 0.7646 0.6211 0.5923 0.4037 0.4056 0.3399 0.2514] |
| 12 | Runtime Limits | Stall Time Limit | value (T) = 4 s | −30.1599 | [0.1119 0.1180 0.1937 0.3267 0.3476 0.4794 0.5589 0.6952 0.7050 0.7677 0.8614 0.7511 0.7524 0.7646 0.6211 0.5923 0.4037 0.4056 0.3399 0.2514] |

**Table A1.** *Cont.*

| Case No. | GA Algorithm Option | Category | Sub-Function Solver Setting | SSL (dB) | Filter Coefficients |
|---|---|---|---|---|---|
| 13 | Tolerances | Function Tolerances | value = $10^{-5}$ | −30.1490 | [0.1114 0.1179 0.1935 0.3265 0.3472 0.4791 0.5588 0.6947 0.7047 0.7677 0.8613 0.7507 0.7521 0.7644 0.6208 0.5923 0.4037 0.4054 0.3399 0.2510] |
| 14 | Tolerances | Function Tolerances | value = $10^{-4}$ | −29.9617 | [0.1085 0.1163 0.1896 0.3215 0.3422 0.4723 0.5575 0.6922 0.7006 0.7659 0.8626 0.7454 0.7475 0.7630 0.6218 0.5917 0.4032 0.4120 0.3399 0.2487] |
| 15 | Tolerances | Fitness Limit | Fitness Limit Value = 0.01 | −30.1599 | [0.1119 0.1180 0.1937 0.3267 0.3476 0.4794 0.5589 0.6952 0.7050 0.7677 0.8614 0.7511 0.7524 0.7646 0.6211 0.5923 0.4037 0.4056 0.3399 0.2514] |
| 16 | Tolerances | Fitness Limit | Fitness Limit Value = 0.05 | −26.3296 | [0.0376 0.0092 0.1892 0.2362 0.3246 0.4347 0.4748 0.6437 0.6288 0.7338 0.8300 0.7065 0.7334 0.7504 0.6110 0.5910 0.3996 0.3229 0.2517 0.3651] |

## Appendix B. Analysis of Random Solutions for Crossover and Mutation

This section compares random seed solutions (no fixed seed) using crossover and mutation functions. For each function, the sub-functions are analyzed, and figures are shown. Four figures show the SLL and how the SLL varies. In each caption, the SLL worst case and best case are specified with mean and median values, followed by the SLL solutions. Figures A1–A4 show the filter's magnitude frequency response with ten solutions for crossover/heuristic, crossover/two-point, mutation adapt-feasible, and mutation/Gaussian, respectively. By analyzing the mean and median values, we observe that mutation/adapt-feasible is the best, followed by mutation/Gaussian, then crossover/two-point, and then the worst case is crossover/heuristic.

Figure A1 depicts ten solutions for the crossover/heuristic sub-settings. The worst case is SLL: −19.0941 dB, and the best case is SLL: −25.5974 dB. Mean = −21.7458 dB. Median = −21.8616 dB. Ten SLL solutions = [−19.0941, −22.1809, −25.5974, −22.0978, −20.8282, −21.9670, −21.7562 −19.2285, −21.4284, −23.2790].

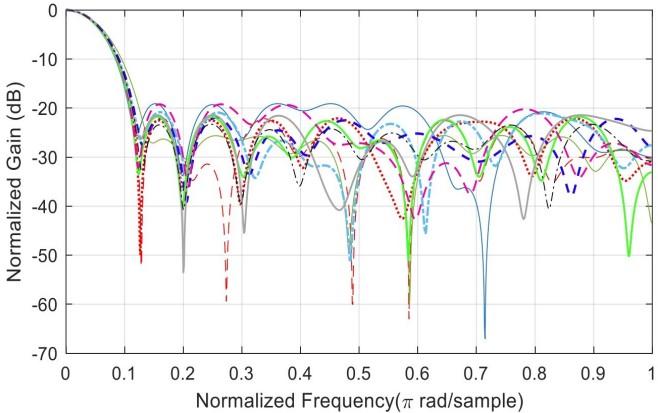

**Figure A1.** Minimization of the SLL. Different colors resembles ten solutions for crossover/heuristic.

Figure A2 shows the minimization of the SLL. Different colors resemble ten solutions for crossover/two-point. Worst case SLL: −28.8705 dB. Best case SLL: −31.7864 dB. Mean = −30.8800 dB. Median = −30.9479 dB. Ten SLL solutions = [−31.1692, −31.7864, −31.5800, −30.6253, −31.3180, −30.6638, −28.8705, −30.7265, −31.6496, −30.4102].

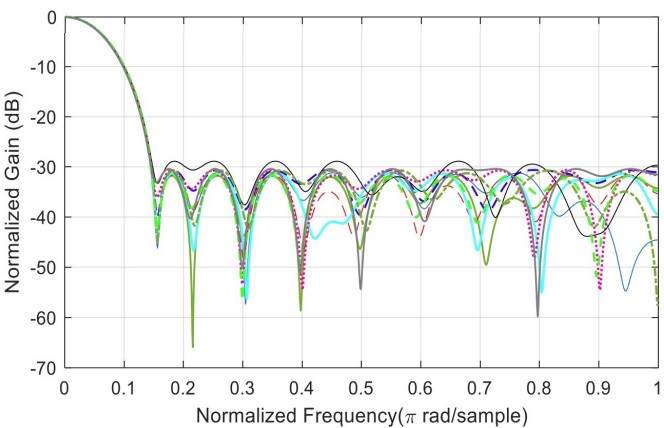

**Figure A2.** Minimization of the SLL. Different colors resembles ten solutions for crossover/two-point.

Figure A3 depicts the minimization of the SLL. Ten (10) solutions for mutation/adapt-feasible. Worst case SLL: −30.7847 dB. Best case SLL: −31.7460 dB. Mean = −31.3768 dB. Median = −31.3983 dB. Ten SLL solutions = [−31.2148, −31.1336, −30.7847, −31.5693, −31.6100, −31.6331, −31.4939, −31.3027, −31.2802, −31.7460].

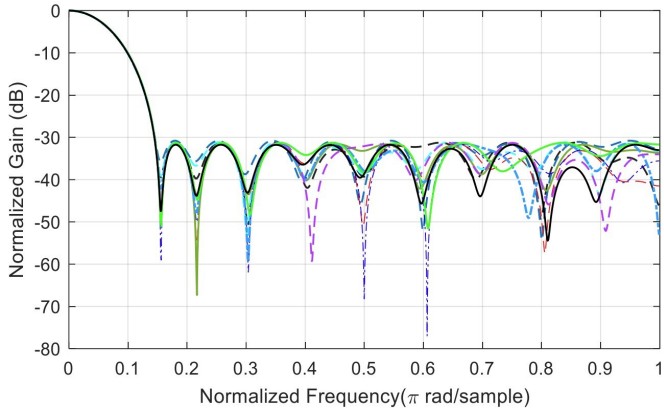

**Figure A3.** Minimization of the SLL. Different colors resembles ten solutions for mutation/adapt-feasible.

Figure A4 displays the minimization of the SLL. Ten (10) solutions for mutation/Gaussian. Worst case SLL: $-29.1817$ dB. Best case SLL: $-32.0514$ dB. Mean = $-30.9744$ dB. Median = $-31.2508$ dB. Ten SLL solutions = [$-29.8972$, $-31.7044$, $-31.2590$, $-31.4947$, $-29.1817$, $-31.2425$, $-31.0011$, $-32.0514$, $-31.8377$, $-30.0746$].

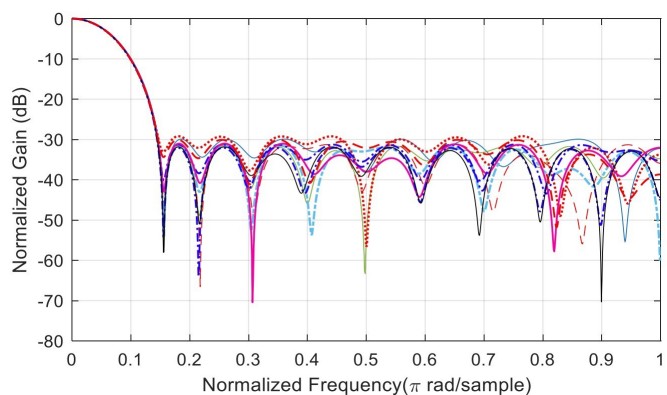

**Figure A4.** Minimization of the SLL. Different colors resembles ten solutions for mutation/Gaussian.

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
