# Peer review of "Optimized FIR Filter Using Genetic Algorithms: A Case Study of ECG Signals Filter Optimization"

_biomedinformatics, doi:10.3390/biomedinformatics3040071_

Round 1

Reviewer 1 Report

Comments and Suggestions for Authors

author proposed an Optimized FIR Filter using Genetic Algorithms.

my remarks to improve the presentation of this paper:

1/ abstract: more detailed the problem and the objective thus the results obtained.

2/ introduction: be more careful when writing each paragraph and especially the transition between ideas, I sometimes find myself lost between paragraphs

3/ related work: very short; must be more detailed and add more work and add a comparative study between the work proposed for optimization;

4/ describe stage two (genetic algorithm) more because this is exactly your contribution; so tried to give value to this part:

5/ question: why did you choose genetic algorithm and not another optimization algorithm like PSO for example

6/ conclusion: add future work to improve your approach

Reviewer 2 Report

Comments and Suggestions for Authors

Overall, the paper presents an interesting approach to optimizing Finite Impulse Response (FIR) filters for electrocardiogram (ECG) biomedical signal noise removal using genetic algorithms. The following are comments and suggestions for improvement:

1. **Clarity and Structure**:

   - The paper is generally well-structured, with clear section headings and a logical flow. However, the introduction could benefit from a more concise and focused presentation of the research problem and objectives.

2. **Motivation and Significance**:

   - The introduction briefly mentions the advantages of genetic algorithms but could emphasize more why optimizing FIR filters for ECG signals is crucial in biomedical engineering. Provide concrete examples of how improved filters can impact the diagnosis of heart conditions.

3. **Abstract**:

   - The abstract is well-written and informative. It effectively summarizes the key contributions and findings of the paper. 

4. **Related Work**:

   - The related work section is informative but lacks a critical analysis of the existing literature. Discuss the strengths and weaknesses of the previous approaches and how your method improves upon them.

5. **Methodology**:

   - The methodology section provides a clear overview of the research process. However, it would be beneficial to provide a brief rationale for choosing a genetic algorithm over other optimization techniques.

   - The description of Stage One (FIR filter design) is concise and informative. Including a short mathematical explanation of the equations and variables used would be helpful for readers less familiar with signal processing.

   - The description of Stage Two (Map and process FIR coefficients using GA) is clear, but it might be useful to explain why 20 chromosomes with 20 genes each were chosen as initial conditions. Discuss how these choices impact the optimization process.

   - Stage Three (Predict the best solution) is adequately described, but it could benefit from a more detailed explanation of the fitness function used to evaluate solutions.

   - Stage Four (Evaluate the best solution) is well-explained. However, clarify how randomness in the GA process can affect the final selection of the best solution.

6. **Figures**:

   - The figures included are relevant and contribute to the understanding of the methodology. However, it would be helpful to label the axes in Fig. 1, Fig. 4, and Fig. 5 for clarity.

7. **Results and Discussion**:

   - The paper mentions that detailed results and analysis are presented in Sections 4.1 to 4.4, but these sections are not included in the provided text. Ensure that these sections are included in the final paper and provide a summary of the key findings in the discussion.

8. **Conclusion**:

   - The abstract mentions "Sec. 6 concludes the paper," but the actual conclusion is not present in the provided text. Ensure that the conclusion section summarizes the main contributions and implications of the research.

9. **References**:

   - The paper references several related works, which is good. However, make sure to include these references properly formatted in the final version of the paper.

10. **Proofreading**:

    - Review the paper for grammar and language issues to ensure clarity and readability.

In summary, the paper presents a valuable approach to optimizing FIR filters for ECG signal processing using genetic algorithms. Addressing the comments and suggestions above will help enhance the clarity and completeness of the paper for readers and researchers in the field of biomedical signal processing and optimization.

Comments on the Quality of English Language

The quality of English language in the paper is generally good, but there are some areas where improvements can be made for clarity and readability. Here are some specific comments on the quality of English language:

1. **Sentence Structure and Clarity**:

   - The paper often uses long sentences that can be complex to follow. Consider breaking some of these sentences into smaller, more digestible sentences to improve readability.

   - In some sentences, there are multiple clauses that may require additional punctuation or restructuring for clarity.

2. **Technical Terminology**:

   - The paper uses technical terminology related to signal processing and genetic algorithms, which is appropriate for the subject matter. However, it's important to ensure that these terms are defined or explained when first introduced to help readers who may not be experts in the field.

3. **Verb Tense**:

   - Be consistent with verb tense throughout the paper. For example, in the abstract, it starts with "The advancement of technology and the availability..." (present tense) but later shifts to "The results of the comprehensive analysis..." (past tense). Maintain a consistent tense throughout.

4. **Acronyms and Abbreviations**:

   - When introducing acronyms or abbreviations, make sure to provide the full term followed by the abbreviation in parentheses the first time they are used. For example, "Finite Impulse Response (FIR) filters" or "side lobe level (SLL)."

5. **Punctuation and Formatting**:

   - Pay attention to punctuation, especially with commas and periods, to ensure that sentences are properly structured.

   - Figure captions and labels should be consistently formatted, and axes in graphs should be clearly labeled.

6. **Conciseness**:

   - Some sentences or phrases can be made more concise. For instance, instead of "The genetic algorithm works by iteratively generating new populations of filters, each of which is a slight variation of the previous population," you can say, "The genetic algorithm iteratively generates new filter populations with slight variations."

7. **Citations**:

   - Ensure that citations are properly formatted and consistently styled throughout the paper.

8. **Consistency**:

   - Ensure that terms and phrases are used consistently throughout the paper. For example, consider whether to consistently refer to "genetic algorithms" or "GAs" and use the chosen term consistently.

9. **Proofreading**:

   - Carefully proofread the paper for grammatical errors and typos. Even small errors can affect the overall readability and professionalism of the paper.

Overall, the paper's English language quality is reasonable, but it can be improved with careful editing and attention to the points mentioned above. Clarity and consistency are key to ensuring that readers can easily understand and follow the research presented in the paper.

Reviewer 3 Report

Comments and Suggestions for Authors

· The authors obtained good results which concern the Filter Coefficients.

·     The introduction needs more improvement and gives the comparison with another method and explains the disadvantages and advantages of these methods used in this paper 

·      The figures results need more explanation.

·       The authors obtained good results which concern the Filter Coefficients.

·       The GENETIC ALGORITHM method gives good results but the denoising method used in this paper needs quality and quantity evaluations. On one hand, the authors used the noisy ECG (Figure 15. Example of one second of the noisy ECG signal) signal without giving the real signal without noise; on the other hand, the evaluation must be improved with other methods such as EMD, EEMD, and CEEMDAN.

·       The authors can use the denoising methods in this reference. CITED ABOVE

[1] Elouaham, S., A. Dliou, Mostafa Laaboubi, R. Latif, N. Elkamoun, and H. Zougagh. "Filtering and analyzing normal and abnormal electromyogram signals." Indonesian Journal of Electrical Engineering and Computer Science 20, no. 1 (October 1, 2020): 176. http://dx.doi.org/10.11591/ijeecs.v20.i1.pp176-184.

[2] Dliou, A.Elouaham, S.Latif, R., M. Laaboubi, Zougagh, H.Saddik, A., Denoising Ventricular tachyarrhythmia Signal. (2018) 9th International Symposium on Signal, Image, Video, and Communications, ISIVC 2018 - Proceedings, art. no. 8709201, pp. 124-128.

·       The authors used in paper this frequency technique to give the magnitude spectrum of the ECG noise signal without giving the name of this method and classically this method can’t give the important component frequency of the ECG signal during the time.

·       The authors can use the time-frequency methods in this reference. CITED ABOVE

[3]   S. Elouaham, R. Latif, A. Dliou, F. Maoulainine, and M. Laaboubi, "Biomedical signals analysis using time-frequency," 2012 IEEE International Conference on Complex Systems (ICCS), Agadir, Morocco, 2012, pp. 1-6, doi: 10.1109/ICoCS.2012.6458575.

·     The results need more interpretation and the figures' results need more explanation.

·     The length of the REAL AND DENOISED signals must be 10 seconds for more quality.

Round 2

Reviewer 1 Report

Comments and Suggestions for Authors

thank you for all your effort in responding to my comments;

before the final version try to clarify equation 2 and 3

Reviewer 3 Report

Comments and Suggestions for Authors

The paper presents the modifications required